# Comparison of Productivity and Cost between Two Integrated Harvesting Systems in South Korea

**Min-Jae Cho [1], Yun-Sung Choi [1], Seung-Ho Paik [1], Ho-Seong Mun [1], Du-Song Cha [2], Sang-Kyun Han [3] and Jae-Heun Oh [1],***

[1]  Forest Technology and Management Research Center, National Institute of Forest Science, 498 Gwangneungsumogwon-ro, Soheul-eup, Pocheon 11186, Korea

[2]  Department of Forest management, Kangwon National University, 1 Gwangwondaehakgil, Chuncheon 24341, Korea

[3]  Department of Forestry, Korea National College of Agriculture and Fisheries, 1515 Kongjwipatwji-ro, Deokjin-gu, Jeonju 54874, Korea

*  Correspondence: jhoh7038@korea.kr; Tel.: +82-31-540-1181

**Abstract:** Interest in the production of renewable energy using forest biomass is increasing in South Korea, and improved knowledge on operations logistics to lower biomass harvesting costs is needed. This study aimed to build a low-cost forest-biomass harvesting system by analyzing the costs of two integrated (cut-to-length and whole-tree) harvesting systems for logs and logging residues. Two integrated harvesting systems were carried out in the clear-cut mixed forest on a steep slope. Compared to the cut-to-length system that separately extracts logs and logging residues in a forest, the cable whole-tree harvesting system can save $8.8/green weight ton (Gwt) because it requires no additional yarding operation cost of logging residues. Moreover, a breakeven analysis shows that the required machine utilization rates that favor whole-tree harvesting systems over cut-to-length harvesting were more than 70% for cable harvesting systems. The introduction of the whole-tree harvesting system is, therefore, required to produce forest biomass at a low cost. In the future, studies on forest-biomass processing and transportation systems will be needed to provide a biomass feedstock supply cost from stump to biomass power plant.

**Keywords:** forest biomass; cut-to-length; whole-tree; integrated harvesting

## 1. Introduction

The concentration of greenhouse gases, such as carbon dioxide, is rapidly increasing globally due to economic activities that use fossil-based fuels. According to the Intergovernmental Panel on Climate Change (IPCC), an increase in greenhouse gas emissions causes climate change, such as global warming [1]. The Environmental Protection Agency (EPA) also introduced the concept of the carbon-neutral effect, in which carbon dioxide is not emitted to the atmosphere if forest biomass is used as a supplied fuel. Therefore, interest in the production of renewable energy using forest biomass instead of fossil fuels is increasing [2,3]. In South Korea, the demand for forest biomass (e.g., wood chips and pellets) is expanding and diversifying to cogeneration and heating for rural areas since the declaration of national strategies for low carbon and green growth in 2008. Because of the implementation of the Renewable Portfolio Standard (RPS) in 2012, large-scale power producers (over 500 MW) are required to use 2% renewable energy sources for power generation, and they are required to increase this to 10% by 2023 [4]. In the forest industry, wood chip manufacturers have gradually increased their purchase of logs from 916,251 $m^3$ in 2012 to 1,431,648 $m^3$ in 2017 [5]. The Donghae Thermal Power Plant used 400,000 t/year of waste wood for dual-fuel power generation

in 2012 and has used only biomass for power generation since 2013. In the future, power plants, paper manufacturers, and public corporations will consistently expand their energy projects to reduce greenhouse gas emissions and secure carbon credits.

Moreover, the Ministry of Trade, Industry and Energy (MOTIE) revised and announced the RPS and Renewable Fuel Standard (RFS) Management and Operation Guidelines in June 2018 for the extraction and utilization of unused forest biomass. Accordingly, the Renewable Energy Certificate (REC) for unused forest biomass was revised to dual-fuel power generation REC 1.5 and biomass power generation REC 2.0 [6]. Therefore, because the flow of the forest market is changing now that forest biomass that was abandoned in forests due to the high cost of extraction is trading at $61.8–63.6/t, studies on low-cost extraction methods for unused forest biomass and its supply systems are needed [7].

For forest-biomass harvesting in South Korea, however, harvesting systems that meet the field conditions have not been applied due to the lack of high-performance forest machines and skilled operators. Private forests accounted for 67.1% of total forests in 2015; the cut-to-length harvesting system, which uses an excavator with a grapple, is mostly applied to private forests for harvesting logs in South Korea [8]. The logging residues that remain after logging are then abandoned in forests. The logging residues reduce planting space, incur post-management costs, and increase the risk of disasters such as forest fires [9]. Therefore, research is needed to efficiently collect logging residues for energy production and hazardous-fuel reduction in the forest.

In North America and Europe, many studies have shown that an integrated harvesting system is defined as a single-pass harvesting operation for maximizing the wood value by collecting logs and logging residues together and using forest biomass as an energy source [10–15]. Integrated harvesting systems produce logs and remove sub-merchantable logging residues for the risk factor of forest fires. Among various harvesting methods, whole-tree harvesting is reported as the most economic method for forest-biomass harvesting [16–18].

Considering the increasing market value for unused forest biomass, it is necessary for South Korea to construct efficient forest-biomass harvesting and supply systems. Many studies have analyzed the productivity and costs of extracting logs using various tree-harvesting systems [19–25]. Although Lee et al. [26] analyzed the productivity and cost factors for forest-biomass harvesting using the cut-to-length harvesting system, studies on the effects of forest-biomass harvesting systems with whole-tree harvesting are still insufficient.

Therefore, in this study, the overall objective was to apply the integrated harvesting system for maximizing the wood value in South Korea. In particular, this study sought to do two things: (1) analyze the productivity and cost of the cut-to-length (ground-based) and whole-tree (cable) integrated harvesting systems on a steep slope, and then (2) compare the systems to the cut-to-length system that separately extracts logs and logging residues. The expected study outcome should help develop integrated harvesting systems that allow for an increased utilization of forest-biomass resources at low costs.

## 2. Materials and Methods

### 2.1. Study Area

Integrated harvesting systems were used in the clear-cut mixed forest (pine 55%, broadleaf trees 40%, and oak 5%) located at 41, Maegok-ri, Hobeop-myeon, Icheon-si, Gyeonggi-do (37°11´36″ N and 127°23´21″ E). Integrated harvesting was performed using cut-to-length (ground-based) and whole-tree (cable) systems (Figure 1).

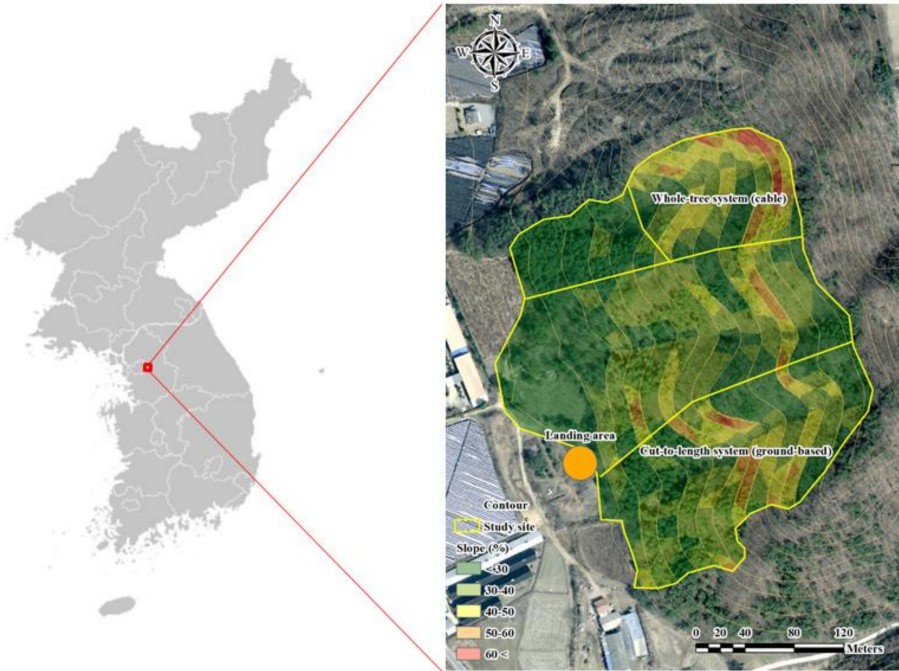

**Figure 1.** Study site.

The area (ha) and slope (%) of the cut-to-length harvesting area were 2.3 ha and 48.2%, and the area and slope of the whole-tree harvesting area were 1.2 ha and 43.3% for the cable system. The stand and site conditions are shown in Table 1.

**Table 1.** Stand descriptions of the study site by integrated harvesting system.

| Items | Integrated Harvesting System | |
|---|---|---|
| | **Cut-to-Length** | **Whole-Tree** |
| | **Ground-Based** | **Cable** |
| area (ha) | 2.3 | 1.2 |
| silvicultural system | clear cut | |
| forest type | mixed forest | |
| species | *Pinus rigida*, *Pinus densiflora*, *Quercus mongolica*, etc. | |
| average slope (%) | 48.2 | 43.4 |
| DBH [a] (cm) | 22/10~48 [b] | 24/8~45 [b] |
| height (m) | 14/10~21 [b] | 14/8~23 [b] |
| stand stock (m³/ha) | 131.1 | 127.5 |

[a] DBH: Diameter at breast height. [b] Numerator means average and denominator means minimum (L) and maximum (R).

### 2.2. Integrated Harvesting System

Forest biomass is defined as products produced in forests [27]. In this study, forest biomass refers to produced logs and logging residues (i.e., tops, branches, and leaves). The integrated harvesting systems, defined as systems that perform forwarding operations for logs and logging residues from the site of felling operations to a landing area, were classified into cut-to-length (ground-based) and whole-tree (cable) harvesting. In South Korea, the cut-to-length integrated harvesting system was preferred not only for gentle slopes but also for steep slopes because of the lack of high-performance forest machines and skilled operators [28]. The cut-to-length integrated harvesting system included a yarding operation that used an excavator (DX55MT-5K DOOSAN, Seoul, Korea) with a grapple, a forwarding operation that used the excavator with a grapple and a crawler-type forest tractor (MST 800VD MOROOKA, Ibaraki, Japan) after felling, and a bucking operation that used a chain saw (MS261

STIHL, Waiblingen, Germany) on steep slopes. For whole-tree integrated harvesting, the felling operation was performed with a chain saw, followed by the yarding operation with a small swing yarder (DX55MT-5K DOOSAN, Seoul, Korea and SW200 HSM, Busan, Korea), the bucking operation with a processor (DX140LCR-5 DOOSAN, Seoul, Korea and 25SH KESLA, Appenweier, Germany), and the forwarding operation with an excavator with a grapple and a crawler-type forest tractor for steep slopes (Figure 2).

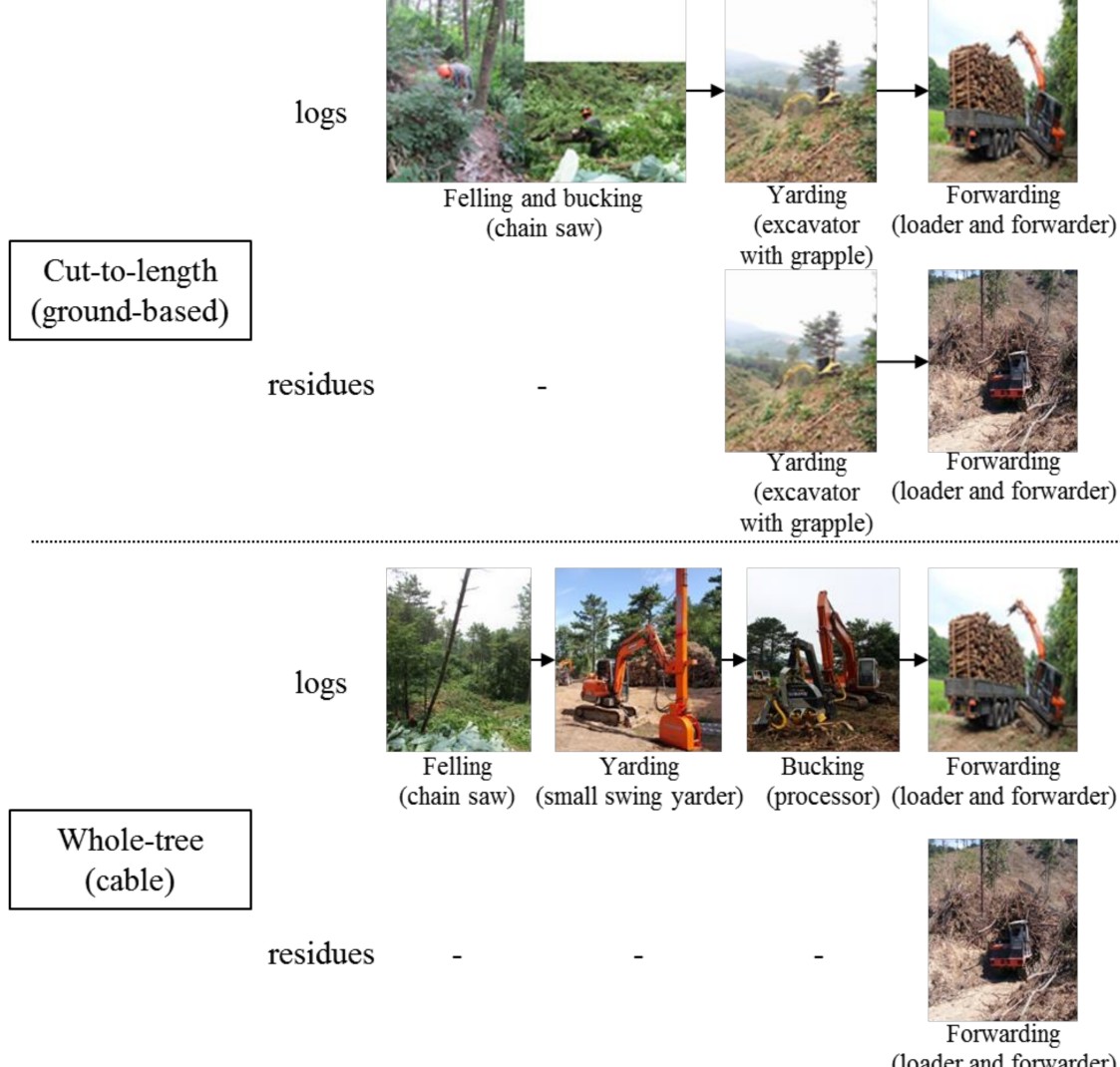

**Figure 2.** Integrated harvesting system.

*2.3. Survey Method*

To calculate the productivity of each harvesting system, the green weight ton (Gwt) was surveyed based on the scheduled machine hour (SMH; day). Each operation element was measured using a stopwatch and the total diameter of the tree. The scale (CAS RW-10L) was used to examine the total weight and the empty weight of the forwarder and truck to calculate the amount of forest biomass. The cost factors of the forest machines used (e.g., initial input cost, fuel consumption, fuel cost, and labor cost), as well as wages for workers, were used to calculate the work cost of the forest-biomass harvesting systems [21–23,29]. Moreover, the stand stock (m³/ha), bucking rate (%), and proportion of each tree species were surveyed to analyze the economic benefits of each harvesting system [30].

*2.4. Analysis Method*

2.4.1. Operation Cost

The operation cost ($/Gwt) was calculated using the machine cost ($/day) and productivity (Gwt/day) as shown in Equation (1):

$$\text{Operation cost (\$/Gwt)} = \frac{\text{Machine cost (\$/day)}}{\text{Productivity (Gwt/day)}}.\tag{1}$$

Gwt and SMH (day) were used to represent productivity (Gwt/day). In order to use the same unit as productivity (Gwt/day) for logs and logging residues, the volume–unit productivity of logs (m$^3$/day) was converted into the weight–unit productivity (Gwt/day) by applying the green wood proportion of each species as shown in Table 2 [30].

**Table 2.** Specific gravity of green wood by species.

|  | *P. rigida* | *P. densiflora* | *Q. variabilis* | *Q. mongolica* | *Q. acutissima* | **Average** |
|---|---|---|---|---|---|---|
| Specific gravity | 0.71 | 0.7 | 1.08 | 1.09 | 1.05 | 0.93 |

The machine cost ($/day) was calculated using the cost of depreciation, interest, fuel, labor, and repair and maintenance, as per the method determined by the Kuratorium für Waldarbeit und Forsttechnik e.V. (KWF) of Germany, as shown in Table 3 [31,32].

Table 3. Cost factors and assumptions used for machine cost calculation by the KWF method.

| Cost Factor | | Unit | Machine | | | | | | |
| | | | | | | Processor | | Small Swing Yarder | |
| | | | Chain Saw | Excavator with Grapple | Forwarder | Excavator | Head | Excavator | Tower Yarder |
|---|---|---|---|---|---|---|---|---|---|
| purchase price | (P) | $ | 818.18 | 49,090.91 | 100,000 | 90,909.09 | 100,000 | 49,090.91 | 65,181.82 |
| endurance period | (N) | years | 1 | 7 | 10 | 7 | 8 | 7 | 7 |
| economic life | (H) | h | 1392 | 9744 | 13,920 | 9744 | 11,136 | 9744 | 14,000 |
| annual operating time a | (J) | h/year | 1392 | 1392 | 1392 | 1392 | 1392 | 1392 | 1392 |
| fuel consumption | (c) | L/h | 0.8 | 8.1 | 6.9 | 22 | - | 16 | - |
| fuel price | (p) | $/L | 1.4 | 1.2 | 1.2 | 1.2 | - | 1.2 | - |
| repair and maintenance | (r) | % | 80 | 80 | 90 | 80 | 90 | 80 | 70 |
| coefficient of lubricant | (l) | % | 50 | 40 | 40 | 40 | 40 | 40 | 40 |
| interest rate | (i) | %/year | 10 | 10 | 10 | 10 | 10 | 10 | 10 |
| machine cost | | | | | | | | | |
| depreciation | P/H or P/(N·H) | $/h | 0.59 | 3.51 | 7.18 | 6.49 | 8.98 | 3.51 | 6.69 |
| interest | 0.5·P·i·0.01/J | $/h | 0.03 | 1.23 | 3.59 | 2.27 | 3.59 | 1.76 | 2.34 |
| repair and maintenance | P/H·r or P·r/(N·H) | $/h | 0.47 | 2.81 | 6.47 | 5.19 | 8.08 | 2.81 | 4.68 |
| fuel price | c·p·(1+l) | $/h | 1.46 | 13.87 | 11.73 | 37.52 | - | 13.87 | - |
| other costs (Insurance, storage fee etc.) | - | $/h | 0.09 | 0.49 | 1.08 | 0.97 | 1.35 | 0.49 | 1 |
| sub total | | $/h | 2.64 | 22.43 | 30.04 | 52.46 | 22 | 22.43 | 14.72 |
| labor cost b (50% inclusion incidental expense) | | $/h | 29.23 | 22.14 | 22.14 | 22.14 | - | 40.1 | - |
| total machine cost | | $/h | 31.87 | 44.57 | 52.18 | 74.60 | 22 | 62.53 | 14.72 |
| | | | | | | 96.60 | | 77.25 | |

a 174 days × 8 h = 1392 h [33]. b Standard of labor cost: feller ($118.18), special worker ($98.40), and forest worker ($79.82) [34].

### 2.4.2. Analysis of the Cost Benefits of the Whole-Tree Integrated Harvesting System

The cost benefits of the whole-tree (cable) integrated harvesting system were calculated based on the cost of the cut-to-length (ground-based) integrated harvesting system.

Moreover, to prepare criteria for introducing whole-tree integrated harvesting systems to forest-biomass harvesting, the productivity (t/h) and cost ($/t) of each harvesting system were calculated according to the machine utilization rate (%), using the scheduled machine hour (SMH) and the productive machine hour (PMH), as shown in Equation (2). The appropriate machine utilization rate (%) of the whole-tree harvesting system was then analyzed based on the cost of the cut-to-length harvesting system ($/Gwt):

$$\text{Machine utilization rate } (\%) = \frac{\text{PMH}}{\text{SMH}} \times 100. \tag{2}$$

## 3. Results and Discussion

### 3.1. Productivity and Costs of the Integrated Harvesting Systems

#### 3.1.1. Cut-to-Length (Ground-Based) Harvesting System

In the forest-biomass cut-to-length harvesting system, the forwarding operation (93.6 Gwt/day per person) exhibited the highest log harvesting system productivity, followed by the yarding operation (58.2 Gwt/day per person) and the felling and bucking operation (20.8 Gwt/day per person). The log harvesting system cost was $24.9/Gwt. The felling and bucking operation had the highest cost ($14.3/Gwt), while the forwarding operation had the lowest ($4.1/Gwt). The logging-residue harvesting system cost was $17.4/Gwt, while the yarding operation cost was $8.4/Gwt, and the forwarding operation cost was $9/Gwt (Table 4).

**Table 4.** Productivity and cost of logs and logging residues production in cut-to-length system.

| System | | | Felling and Bucking (Chain Saw) | Yarding (Excavator with Grapple) | Forwarding a (Excavator with Grapple + Forwarder) | Total |
|---|---|---|---|---|---|---|
| Cut-to-length | logs | productivity (green weight ton (Gwt)/day·man) | 20.8 | 58.2 | 93.6 | - |
| | | cost ($/Gwt) | 14.3 | 6.5 | 4.1 | 24.9 |
| | | machine utilization rate (%) | 54.4 | 85.7 | 87.1 | - |
| | Logging residues | productivity (Gwt/day·man) | - | 42.4 | 43.2 | - |
| | | cost ($/Gwt) | - | 8.4 | 9 | 17.4 |
| | | machine utilization rate (%) | - | 85.7 | 97 | - |

a Forwarding distance: less than 150 m.

#### 3.1.2. Whole-Tree (Cable) Harvesting System

In the forest-biomass whole-tree harvesting system, the bucking operation (131.2 Gwt/day per person) exhibited the highest log harvesting system productivity (Gwt/day per person), followed by felling operation (171.2 Gwt/day per person), forwarding operation (93.6 Gwt/day per person), and yarding operation (24.8 Gwt/day per person). The log harvesting system cost ($/Gwt) was $24.5/Gwt. The yarding operation cost ($12.5/Gwt) was the highest, while the felling operation cost ($3.1/Gwt) was the lowest. As for the logging-residue harvesting cost, the forwarding operation cost of $9/Gwt was incurred due to the use of an excavator with a grapple and a crawler-type forest tractor (Table 5).

**Table 5.** Productivity and cost of logs and logging residues production in whole-tree system.

| System | | Felling (Chain Saw) | Yarding (Small Swing Yarder) | Bucking Processor) | Forwarding [a] (Excavator with Grapple + Forwarder) | Total |
|---|---|---|---|---|---|---|
| Whole-tree (cable) | logs | productivity (Gwt/day·man) 95.2 | 24.8 | 103.2 | 93.6 | - |
| | | cost ($/Gwt) 3.1 | 12.5 | 4.8 | 4.1 | 24.5 |
| | | machine utilization rate (%) 56.9 | 80.4 | 95.3 | 87.1 | - |
| | Logging residues | productivity (Gwt/day·man) - | - | - | 43.2 | - |
| | | cost ($/Gwt) - | - | - | 9 | 9 |
| | | machine utilization rate (%) - | - | - | 97 | - |

[a] Forwarding distance: less than 150 m.

### 3.1.3. Operation Element Cost for Each Integrated Harvesting System

The cut-to-length integrated harvesting system where a chain saw was used exhibited the highest felling and bucking operation cost ($14.3/Gwt), followed by the whole-tree harvesting system that used a chain saw and a processor ($7.9/Gwt). The yarding operation cost for the whole-tree system that used a small swing yarder was $12.5/Gwt, and the yarding operation cost for the cut-to-length harvesting system that used an excavator with a grapple was $14.9/Gwt. The forwarding operation cost was $13.1/Gwt using an excavator with a grapple and a crawler-type forest tractor (Figure 3).

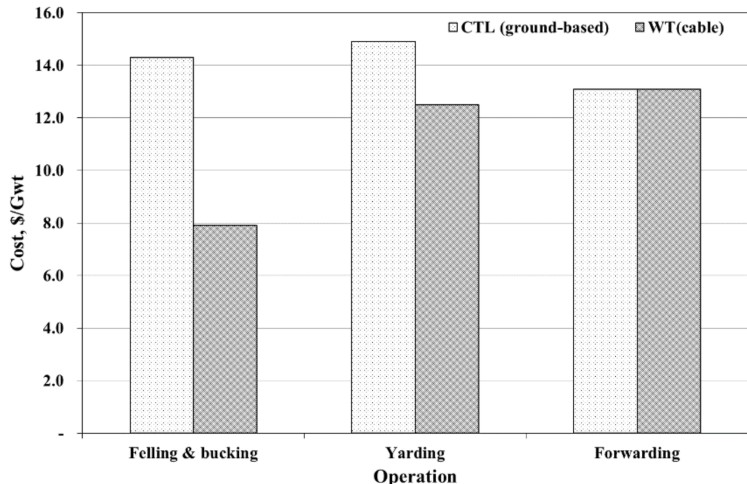

**Figure 3.** Operation cost by integrated harvesting systems.

### 3.2. Analysis of the Cost Benefits of Integrated Harvesting System

### 3.2.1. Cost Reduction of Whole-Tree Harvesting Systems

The cost benefits of the whole-tree system were calculated based on the cost of cut-to-length harvesting system in order to analyze the economic benefits of the whole-tree harvesting system. When the whole-tree system was applied to forest-biomass harvesting, $8.8/Gwt was saved compared to the cut-to-length harvesting system (Table 6).

**Table 6.** Cost of logs and logging residues production by integrated harvesting systems.

| System / Forest Biomass | Cut-to-Length(Ground-Based) | | Whole-Tree(Cable) | |
| --- | --- | --- | --- | --- |
| | Cost ($/Gwt) | Standard | Cost ($/Gwt) | Cost Reduction ($/Gwt) |
| logs | 24.9 | - | 24.5 | 0.4 |
| logging residues | 17.4 | - | 9 | 8.4 |
| total | 42.3 | - | 33.5 | 8.8 |

In a cost analysis of whole-tree and cut-to-length harvesting systems by Kim and Park [20], it was reported that the cost of the cut-to-length harvesting system was lower than that of the whole-tree harvesting system. This was because the cost was high ($24.3/Gwt) for bucking operations using an excavator with a grapple and a chain saw in the whole-tree harvesting system. When the cost ($4.8/Gwt) of bucking using a high-performance processor machine was applied in this study, it was found that the whole-tree harvesting system was more economical than the cut-to-length harvesting system, with total costs of $57.1/Gwt and $61.8/Gwt, respectively (Table 7). It is therefore necessary to introduce the whole-tree system that uses high-performance forest machines for forest-biomass harvesting.

**Table 7.** Comparison of whole-tree (WT) and cut-to-length (CTL) harvesting system cost (unit: $/Gwt).

| Study | System | Felling (Chain Saw) | Yarding (Small Swing Yarder) | | Bucking (Excavator with Grapple + Chain Saw) | Forwarding (Forwarder) | Total |
| --- | --- | --- | --- | --- | --- | --- | --- |
| Kim and Park (2012) | WT | 2.7 | 23.4 | | 24.3 | 26.2 | 76.6 |
| Kim and Park (2012) + this study | | felling (chain saw) | yarding (small swing yarder) | | bucking (processor) | forwarding (forwarder) | |
| | WT | 2.7 | 23.4 | | 4.8 | 26.2 | 57.1 |
| Kim and Park (2012) | | felling and bucking (chain saw) | yarding for logs (excavator with grapple) | yarding for residues (excavator with grapple) | set up of skidding trail (excavator with grapple) | forwarding (forwarder) | |
| | CTL | 13 | 9.7 | 8.9 | 3.9 | 26.2 | 61.8 |

## 3.2.2. Appropriate Machine Utilization Rate for the Introduction of Whole-Tree Integrated Harvesting Systems

The cost ($/Gwt) of each harvesting system was analyzed according to the machine utilization rate. As a result, for the cut-to-length harvesting system, the log harvesting cost ranged from $19.8/Gwt to $178.5/Gwt, while the logging residue harvesting cost ranged from $17.6/Gwt to $158.1/Gwt (Figure 4).

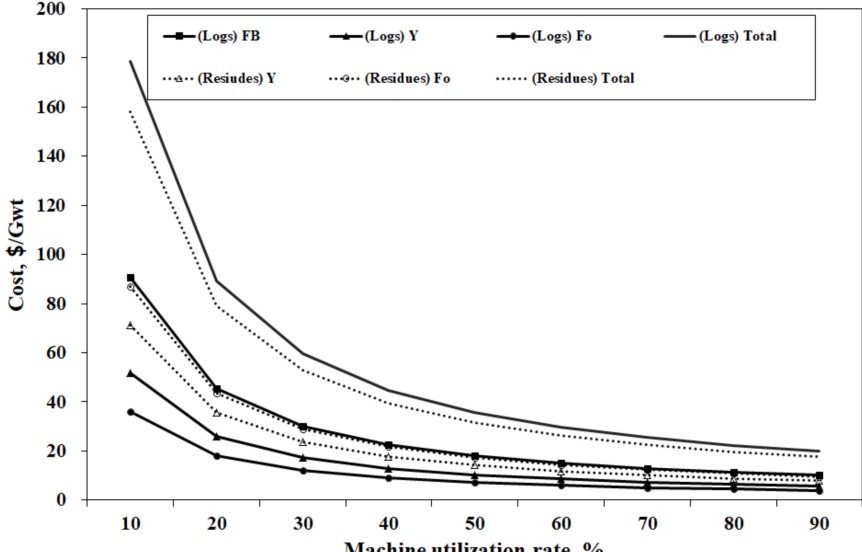

**Figure 4.** Cost of forest-biomass production in cut-to-length harvesting system by machine utilization rate (FB—felling and bucking; Y—yarding; Fo—forwarding).

For the cable whole-tree harvesting systems, the log harvesting costs ranged from $22.5/Gwt to $202.2/Gwt, while the logging residue harvesting cost ranged from $9.7/Gwt to $86.9/Gwt (Figure 5).

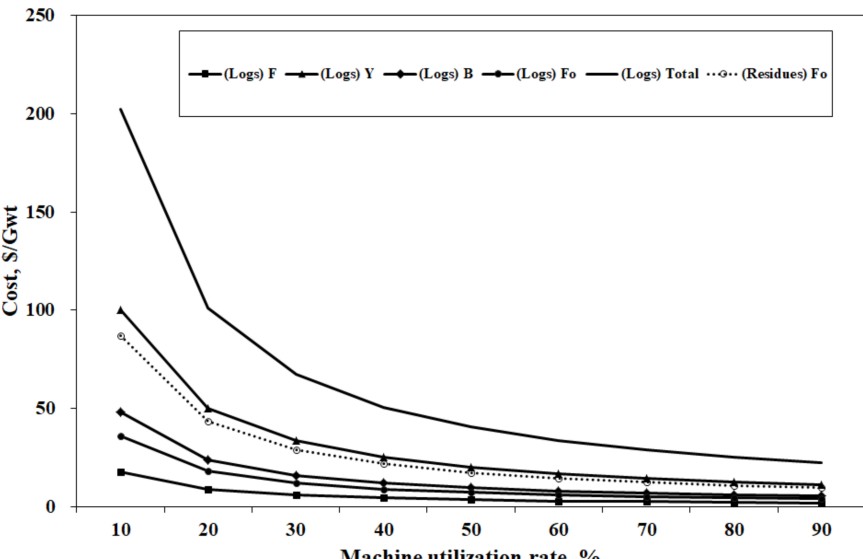

**Figure 5.** Cost of forest-biomass production in whole-tree (cable) harvesting system by machine utilization rate (F—felling; Y—yarding; B—bucking; Fo—forwarding).

To prepare criteria for introducing the whole-tree harvesting systems to forest-biomass harvesting, the appropriate machine utilization rates of the systems were analyzed based on the cost of the cut-to-length harvesting system ($42.0/Gwt). The cable whole-tree harvesting systems were more economical than the cut-to-length harvesting system when the machine utilization rates were higher than 70% (Figure 6). Therefore, it is necessary to improve the machine utilization rate by reducing the delay time in tasks that use high-performance forest machines, such as harvesters, small swing yarders, grapple skidders, and processors, so as to introduce whole-tree harvesting systems for forest-biomass harvesting.

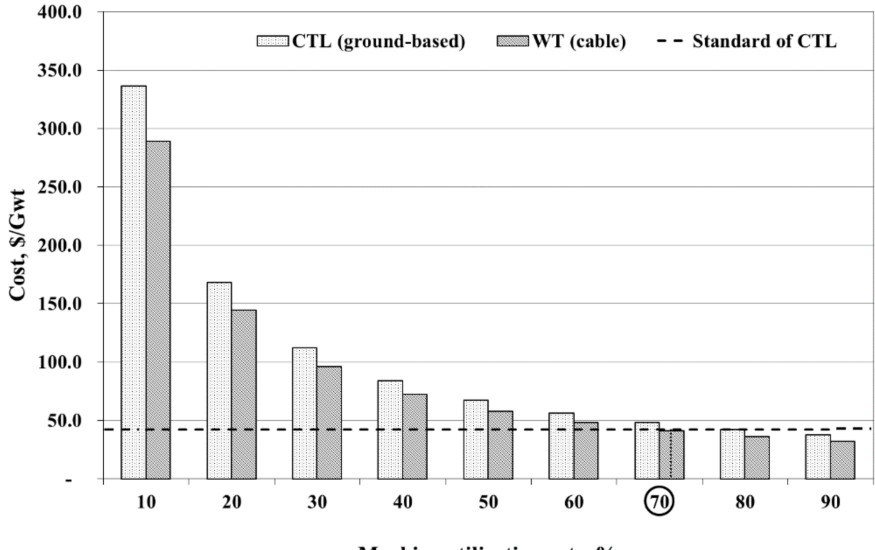

**Figure 6.** Optimal machine utilization of whole-tree harvesting systems by standard of cut-to-length harvesting system cost (CTL—cut-to-length; WT—whole-tree).

## 4. Conclusions

As a result of the efforts to reduce greenhouse gas emissions that cause global warming, interest in producing renewable energy using forest biomass as an alternative to fossil fuels is growing globally. In South Korea, policies for the use of unused forest biomass were prepared as a declaration of national strategies for low carbon and green growth in 2008, the Renewable Portfolio Standard (RPS) in 2012, and the revision of the Renewable Energy Certificate (REC) weight in 2018. This study aimed to construct a low-cost forest-biomass harvesting system by comparing and analyzing the costs of two integrated (cut-to-length and whole-tree) harvesting systems for logs and logging residues.

Compared to the cut-to-length system that separately extracts logs and logging residues in a forest, the cable whole-tree harvesting system saved $8.8/Gwt, because it required no additional yarding operation costs to deal with logging residues.

Moreover, the required appropriate machine utilization rate for the whole-tree harvesting systems was over 70% for the cable system based on the cost of the cut-to-length harvesting system. It is necessary to establish training programs for operators and other supports to improve the machine utilization rates of whole-tree harvesting systems using high-performance forest machines for the production of forest biomass.

For the harvesting of forest biomass at a low cost for a future energy source, whole-tree harvesting systems must be introduced and additional studies on forest-biomass processing and transportation systems must be conducted.

**Author Contributions:** Investigation, Y-S.C., S-H.P. and H-S.M.; writing—original draft preparation, M-J.C.; writing—review and editing, S-K.H. and J-H.O.; supervision, D-S.C.

**Funding:** This research received no external funding.

**Conflicts of Interest:** The authors declare no conflict of interest.

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
