# Peer review of "Comparison of Productivity and Cost between Two Integrated Harvesting Systems in South Korea"

_forests, doi:10.3390/f10090763_

Round 1

Reviewer 1 Report

It would be nice to add a transcript of the abbreviation "DBH" in the table 1  (line 94) and explain what is in the numerator and denominator in the rows "DBH" and "height".
Very poor pictures quality in Figure 2 (line 111). I could not recognize the tractor MOROOKA MST 800VD. The whole-tree method uses a processor for the bucking, but this machine is missing in the table 3 (line 134).
Cost of logs in the table 4 (line 154) is 24.9 $/Gwt, but the same cost in the table 6 (line 183) is 24.5 $/Gwt  

Author Response

Response to Reviewer 1 Comments

Overall answer: We tried to reflect the opinions of the reviewers. Thank you for your kind comments. Modifications were marked in red.

Point 1: It would be nice to add a transcript of the abbreviation "DBH" in the table 1  (line 94) and -explain what is in the numerator and denominator in the rows "DBH" and "height".

 Response 1: We added a transcript of the abbreviation “DHB” and what is in the numerator and denominator in the Table 1 (in red).

Point 2: Very poor pictures quality in Figure 2 (line 111). I could not recognize the tractor MOROOKA MST 800VD. The whole-tree method uses a processor for the bucking, but this machine is missing in the table 3 (line 134).

Response 2: We changed the Figure 2 to reflect the comment of the reviewer and added the processor cost in Table 3 (in red).

Point 3: Cost of logs in the table 4 (line 154) is 24.9 $/Gwt, but the same cost in the table 6 (line 183) is 24.5 $/Gwt.

Response 3: It was the $24.9/Gwt, so we changed the cost of logs in the Table 6 and the cost reduction ($8.8/Gwt) in line 21, 185, and 234 (in red).

Reviewer 2 Report

The article deals with the interesting issue of comparing the costs of forest stands logging on steep slopes, using two logging methods: the cut-to-length method and the whole tree method. Methodologically, the article corresponds to the usual procedures - the methodology is logical and correct. Perhaps I would just recommend specifying the procedure for determining the amount of green biomass produced. The selected areas where the research was conducted are very similar to each other and it can be therefore assumed that the obtained results are not distorted due to different site conditions. Pictures, graphs and tables suitably complement the text of the article. I recommend adjusting Figure 2, which is, in my opinion, less legible and difficult to understand. Among other things, I consider the fact that the authors used the well-known principle according to the German organization KWF when assessing the cost of both technologies. This is one of the reasons why the informations presented in the article are interesting both for researchers and experts in forestry practice. The statements made at the end of the article can be accepted.

In my experience, an important aspect in assessing mining technologies is not only the price at which the work is done, but also the degree of damage to the forest environment by operating machinery, dragging timber, etc.

Author Response

Response to Reviewer 2 Comments

Overall answer: We tried to reflect the opinions of the reviewers. Thank you for your kind comments. Modifications were marked in red.

Point 1: I would just recommend specifying the procedure for determining the amount of green biomass produced.

 Response 1: We added how to determine the amount of green biomass produced to reflect the comment of the reviewer in line 118-120 (in red).

Point 2: I recommend adjusting Figure 2, which is, in my opinion, less legible and difficult to understand.

Response 2: We changed the Figure 2 to reflect the comment of the reviewer (in red).

Point 3: In my experience, an important aspect in assessing mining technologies is not only the price at which the work is done, but also the degree of damage to the forest environment by operating machinery, dragging timber, etc.

Response 3: We agree with your opinion. It is important both the price and the damage of forest environment. In this study, we conducted to compare how different between cut-to-length and whole-tree systems for forest biomass at the price because there was a few studies to compare that in South-Korea. In the future, we will conduct on the damage of the forest environment as well as the price.
